# The Relationship between Telomere Length and Nucleoplasmic Bridges and Severity of Disease in Prostate Cancer Patients

**DOI:** 10.3390/cancers15133351

**Published:** 2023-06-26

**Authors:** Varinderpal S. Dhillon, Permal Deo, Michael Fenech

**Affiliations:** 1Health and Biomedical Innovation, Clinical and Health Sciences, University of South Australia, Adelaide 5000, Australia; permal.deo@unisa.edu.au; 2Genome Health Foundation, North Brighton 5048, Australia

**Keywords:** telomere length (TL), nucleoplasmic bridges, prostate cancer (PC), Gleason score

## Abstract

**Simple Summary:**

Telomeres are essential to prevent telomere end fusions that cause the formation of dicentric chromosomes and nucleoplasmic bridges (NPBs) which in turn induce chromosomal instability, cancer initiation and progression. In this study, we show that telomere length (TL) is reduced and NPBs are increased in prostate cancer (PC) cases and the incidence of these adverse genomic insults is further elevated with increased severity of disease. NPBs and TL in white blood cells may have clinical utility to identify and triage PC cases who may be at risk of serious disease. More studies are required to confirm our observations and to explore mechanistic differences in the role of telomeres in NPB formation in PC cases relative to healthy controls.

**Abstract:**

Telomeres are repetitive nucleotide (TTAGGG) sequences that stabilize the chromosome ends and play an important role in the prevention of cancer initiation and progression. Nucleoplasmic bridges (NPBs) are formed when chromatids remain joined together during mitotic anaphase either due to mis-repair of DNA breaks or due to chromatid end fusion as a result of telomere loss or telomere dysfunction. We tested the hypotheses that (i) telomere length (TL) is shorter in prostate cancer (PC) patients relative to healthy age-matched individuals, (ii) TL differs in different stages of PC and (iii) shorter TL is significantly correlated with NPBs formation in PC cases. TL was measured in whole blood by well-established quantitative PCR method and the frequency of NPBs was measured in lymphocytes using cytokinesis-block micronucleus cytome (CBMNcyt) assay. Our results indicate that TL is shorter and NPBs are increased in PC patients relative to age-matched healthy controls. Furthermore, TL was significantly shorter (*p* = 0.03) in patients with a Gleason score more than 7 and there was also a significant trend of decreasing TL across all three stages (*p* trend = 0.01; Gleason score <7, 7 and >7). Furthermore, TL was significantly inversely correlated with NPB frequency in PC patients (r = −0.316; *p* = 0.001) but not in controls (r = 0.163; *p* = 0.06) and their relationships became stronger with higher Gleason scores. More studies are required that can confirm our observations and explore mechanistic differences in the role of telomeres in NPB formation in PC cases relative to non-cancer cases.

## 1. Introduction

Prostate cancer (PC) is the second most common cancer in men. It is a multi-step process involving accumulation of both genetic and epigenetic aberrations that confer proliferative advantage to pre-cancerous cells [1,2]. The risk of cancer is also affected by various factors such as ethnicity, smoking, family history of cancer, consumption of a high fat diet and alcoholism [3]. Cancer, including PC, causes a significant financial and social burden especially in ageing populations [4]. Understanding the fundamental causes of cancer can identify the most appropriate biomarkers for triaging those most at risk of PC and help identify efficacious preventative diet and lifestyle therapies. In this study, we tested the relevance of biomarkers of telomere dysfunction, i.e., telomere shortening and nucleoplasmic bridges.

Telomeres located at the terminal end of chromosomes are nucleoprotein complexes composed of repeat nucleotide sequences (TTAGGG)n and shelterin proteins [5,6]. They protect the chromosome ends from exonucleolytic degradation that subsequently causes telomere end fusion and abnormal dicentric chromosome formation [7,8,9]. Telomeres are, therefore, essential for the maintenance of genetic integrity, and shortening of telomeres is associated with accelerated ageing and genomic instability [1,10]. Oxidative stress [11] and chronic inflammation [12,13] are the possible mechanisms leading to telomere shortening, and they have been proposed as a cancer risk biomarker in the prostate [9,14]. Many studies have examined the association of lymphocyte telomere length with cancer; however, the results are contrasting [15,16]. Both short and long telomeres have been reported to increase cancer risk including prostate cancer [17,18,19,20,21,22].

Increases in DNA damage biomarkers such as micronuclei (MN) provide valuable information about the defects in DNA repair and chromosome segregation that leads to the chromosome instability phenotype that is commonly observed in cancer [23,24]. In contrast to MN, nucleoplasmic bridges (NPBs) provide an additional measure of mis-repaired DNA breaks as well as defective separation of sister chromatids during anaphase in mitosis due to telomere end-fusion or failure of de-catenation; nuclear buds (NBuds) are caused by the extrusion of amplified DNA and/or unresolved DNA repair complexes [25,26]. Spontaneous frequency of NPBs has been reported to be significantly elevated in cancer, including urothelial cancer and prostate cancer [27,28,29,30,31].

It can be assumed that the mechanisms for the induction of DNA damage may be similar in different tissues and the extent of DNA damage evaluated in lymphocytes and other surrogate tissues is likely reflective of the level of genomic damage in cancer-prone tissues and subsequent cancer risk [26,32]. Telomere–telomere end fusions induced by telomere erosion or deletion in proliferating cells, together with cell cycle checkpoint inactivation, could generate dicentric chromosome anaphase bridges the major chromosomal instability that is responsible for initiating epithelial carcinogenesis in humans [33,34]. Anaphase bridges from which NPBs originate have been closely related to chromosomal instability in human cancers [35]. Therefore, loss of telomere integrity appears to be a critical event in cancer progression. Furthermore, it is important to ascertain if genomic instability mediated by telomere shortening is associated with NPBs; hence, the present study was carried out to test the hypotheses that (i) telomere length (TL) is shorter in PC patients relative to healthy age-matched individuals, (ii) TL differs in different stages of the disease and (iii) shorter TL is significantly correlated with NPBs formation in PC cases.

## 2. Materials and Methods

### 2.1. Study Design and Participants

This is a hospital-based case–control study. The details about the study population, recruitment and inclusion/exclusion criteria are described below and in our previous publications [36,37,38].

The hospital-based case–control study was conducted in South Australia by a collaborative team from Royal Adelaide Hospital and the Commonwealth Scientific and Industrial Research Organisation of Australia (CSIRO). The study design was approved by the Royal Adelaide Hospital and CSIRO research ethics committees. All subjects gave written informed consent for participation. All cases were untreated male Caucasian patients (n = 106) with histologically confirmed prostate cancer. The indication for prostate biopsy was a suspicious finding on digital rectal examination and/or elevated serum levels of prostate-specific antigen (PSA; 0.08–45 ng/mL). The aggressiveness of the prostate tumor of the patient defined by the Gleason score varied between 6 and 9. All patients who were part of this study were classified as requiring radiotherapy for cancer control. The study population of prostate cancer cases was derived among patients referred to the Radiation Oncology Department of the Royal Adelaide Hospital for radical external beam radiotherapy (EBRT) for localized prostate cancer (UICC TNM Stage T1–T3, N0 M0). The selected participants met the eligibility criteria of: (1) suitability for EBRT; and (2) patient consent based on the written protocol approved by the Research Ethics Committees of the Royal Adelaide Hospital and CSIRO (Division of Food and Nutrition).

Age-matched controls (n = 132) were male individuals who did not have any apparent sign of cancer including prostate cancer at the time of recruitment, had normal plasma PSA levels (0.0–3.0 ng/mL) and were not taking any medication for the treatment of life-threatening diseases. Cases and controls in this study were matched with respect to age. We did our best to match smoking status between groups but we were limited by the difficulty in recruiting smokers in the control group. Table 1 describes the characteristics of cases and controls.

### 2.2. Blood Collection and CBMN Cytome Assay

Blood was collected from PC patients and controls after an overnight fast in lithium heparin tubes. An hour after blood collection, 500 μL of whole blood was mixed with 4.5 mL of pre-warmed RPMI-1640 culture medium (Thermo Trace, Melbourne, Australia) supplemented with 10% fetal calf serum (FCS; Thermo Trace, Australia). To induce radiation-induced DNA damage in lymphocytes, the whole blood cultures were exposed to 3 Gy γ-rays from a Cs-137 source (Cis Bio IBL 437 C Blood Product Irradiator, dose rate 5.34 Gy/min). CBMN cytome assay was performed as described previously [39], with slight modifications, and whole blood cultures were set up in duplicate. The detailed protocol and scoring criteria for MN, NPBs and NBuds is explained in our previous publications [37,39].

### 2.3. DNA Isolation and Real-Time qPCR Assay for Measuring TL

Blood was collected in an EDTA tube to isolate DNA, and Genomic DNA was extracted using the QIAamp DNA blood mini kit (Qiagen, Clayton, Australia). An OD ratio (1.8–2.0) of 260:280 was used an indicator of DNA purity. Purified DNA samples were quantified using NanoDrop 1000 spectrophotometer (Thermo Fisher Scientific, Thebarton, Australia) and diluted as per experimental requirements (5 ng/μL). Telomere length was measured using quantitative real-time PCR, as described previously [40,41]. The detailed protocol is explained in detail in our previous publication [41].

### 2.4. Statistical Analysis

All data parameters were analyzed for Gaussian distribution to determine whether to use parametric or non-parametric tests. To determine significance of the differences between the two groups with regards to age, PSA levels, baseline and radiation-induced NPBs, MN and NBuds in bi-nucleated cells and telomere length, we used the unpaired non-parametric Student’s *t* test. Correlation analysis was performed by Spearman’s or Pearson’s test, depending on whether the biomarker data were Gaussian or non-Gaussian in their distribution. All analyses were performed using PRISM 9.0.0 (GraphPad software, San Diego, CA, USA), and *p* values < 0.05 were considered statistically significant.

## 3. Results

### 3.1. Demographic and Clinical Characteristics

The demographic and clinical characteristics of the 106 cases and 132 controls are summarized in Table 1. Cases and controls did not differ significantly in terms of age. PC cases were 2.14 years older (mean age 71.24 ± 7.18) than the controls (mean age 69.07 ± 7.99). PC cases had a 4-fold greater plasma PSA concentration compared to the controls (*p* = 0.0001). TL in PC cases was marginally shorter than healthy age-matched controls (*p* = 0.26). The distribution of PC cases with regard to Gleason score (<7, 7 and >7) was 23, 42 and 41, respectively. Baseline frequency of NPBs was significantly higher in PC cases compared to healthy controls (4.26 ± 0.28 vs. 2.72 ± 0.18, respectively; *p* < 0.0001). However, radiation-induced frequency of NPBs was not significantly different in PC cases and controls (151 ± 4.27 vs. 149 ± 2.86; *p* = 0.62).

### 3.2. Relationship between TL and NPBs in PC Patients

Baseline or radiation-induced frequency of MN and NBuds show an inverse correlation with TL, but did not reach a statistically significant level (Figure 1A,B,E,F). However, there was a significant inverse correlation between the baseline or radiation-induced frequency of NPBs with TL (r = −0.316 and r = −0.284, respectively) in PC patients (Figure 1C,D).

### 3.3. Relationship between TL, NPBs and Gleason Score in PC Patients

TL was significantly shorter in PC cases with Gleason score (GS) > 7 compared to those with GS < 7 (*p* = 0.03; Figure 2A), and we also found a significant trend in declining TL with respect to increasing GS (*p* trend = 0.01; Figure 2A). However, the frequency of NPBs shows an increasing trend with increasing GS (*p* = 0.04; *p* trend = 0.01; Figure 2B). In addition, there is also a significantly increasing trend in radiation-induced NPB frequency with increasing GS (*p* trend = 0.05; Figure 2C). There was a non-significant inverse correlation between baseline or radiation-induced NPB frequency with TL in PC patients with a Gleason score of less than 7 (r = −0.356; *p* = 0.1; r = −0.234; *p* = 0.29, respectively; Figure 3A,B). However, there was a significant inverse correlation between baseline or radiation-induced frequency of NPBs and TL in PC patients with a Gleason score of 7 (r = −0.397; *p* = 0.01; r = −0.339; *p* = 0.03, respectively; Figure 3C,D). In PC patients with a Gleason score of > 7, the inverse correlation was highly significant between baseline or radiation-induced NPB frequency with TL (r = −0.404; *p* = 0.008; r = −0.403; *p* = 0.008, respectively; Figure 3E,F).

Overall, the strength and statistical significance of the correlation of NPBs with TL increased with a higher GS score.

Furthermore, we also observed that PSA is significantly inversely correlated with TL (r = −0.337, *p* = 0.0004) and positively correlated with NPBs (r = 0.249, *p* = 0.01) in PC cases (Figure 4A,B). However, in the controls, TL or NPBs were not significantly correlated with PSA (Figure 4C,D), possibly due to their narrower range of PSA values.

## 4. Discussion

The results of our study show that TL is short in PC patients compared to controls. Furthermore, TL is significantly shorter in PC patients with Gleason score >7 compared to those cases with a score <7, and also compared to controls. Similarly, it has recently been shown that TL was significantly short in PC patients with higher Gleason scores [20,42]. Therefore, the results obtained in the present study support the hypothesis that short telomere length in leukocytes is a predictor of PC risk and aggressiveness of PC. However, the results from some earlier studies, including a recent meta-analysis, were inconsistent with regard to length of telomeres, aggressiveness of the disease, type of tissue studied, obesity, age, race, poor survival rate and lifestyle factors [15,19,21,43,44,45,46,47]. In addition, there are reports showing an association between shorter TL in prostate cancer tissue compared to normal adjacent tissue [48,49]. Shorter TL in leukocytes may reflect another aspect of prostate cancer risk involving weaker immune function due to low telomerase activity [50].

The frequency of lymphocyte NPBs in many cancers including prostate cancer from our previous report have been shown to be significantly higher compared to the age-matched controls [27,28,29,30,31,51]. These results indicate that NPBs are an important biomarker associated with genomic instability in cancers. However, its relationship with TL in prostate cancer was not previously tested and needs to be further verified in other larger cohorts.

Genomic instability is an evolving hallmark of most cancer cells resulting from mutations in DNA repair genes and exposure to exogenous and endogenous genotoxins, as well as malnutrition. Most cancers exhibit chromosomal instability (CIN) evident by a high rate of chromosomal aberrations and abnormal chromosome number, which changes over time [52,53]. In addition, abnormal nuclear aberrations such as micronuclei (MN), nucleoplasmic bridges (NPBs) and nuclear buds are also associated with cancer cells [54]. Telomere uncapping is one of the mechanisms for the generation of CIN, as shown by the loss of T-loop structures due to telomere attrition, leading to impaired protection of chromosomal ends. This results in the formation of chromosome end fusions and dicentric chromosome formation, which leads to nucleoplasmic bridge formation during anaphase and repeating cycles of chromosome instability by the breakage-fusion-bridge cycle mechanism [55,56]. In addition, the positioning of telomeres within the nucleus is also highly specific and dependent on the telomere interactions with the nuclear envelope either directly or indirectly through chromatin-interacting proteins [57]. Therefore, we investigated if there is any relationship between the frequency of DNA damage biomarkers such as MN, NBuds and especially NPBs with telomere attrition in prostate cancer patients. The frequency of baseline or radiation-induced MN and Nbuds is not significantly associated with TL in PC patients. In contrast, results from the present study indicated that there is a strong inverse correlation between NPBs and telomere length in prostate cancer patients. NPBs are increased in PC patients, whereas MN frequency is not significantly elevated compared to controls. In addition, TL is inversely correlated with NPBs in PC patients, but this is not the case with MN frequency. Therefore, the likely mechanism causing NPB formation is telomere shortening rather than telomere loss because the latter would cause increases in both NPBs and MN, unlike the former, which only generates NPBs (Figure 5). Furthermore, this inverse relationship between NPB and TL is stronger with increasing Gleason scores, which also suggests that the propensity of shorter TL to cause NPB formation and chromosomal instability is substantially increased in those with more severe PC.

The findings of our study suggest that NPBs and TL may have clinical utility to identify men who may be at increased risk of PC initiation, as well as those who are more prone to severe PC progression. These men could then be triaged for more frequent follow-up and treatment rather than watchful waiting. The utility of the NPB and TL biomarkers can be further enhanced if they are also used to identify which diet and lifestyle interventions can substantially reduce telomere shortening and NPB formation in those men who are susceptible to PC.

Although MN were not increased in prostate cancer patients, it is important to ponder whether NPB, like MN, may also be a source of DNA leaking into the cytoplasm and triggering the pro-inflammatory cGAS-STING pathway [58]. To the best of our knowledge, there is no convincing evidence yet that NPBs directly activate the cGAS-STING mechanism. However, it has been reported that when NPBs break, they may result in the formation of MN or NBUDs, which in turn might lead to DNA leakage into cytoplasm if their nuclear membranes are disrupted [58,59].

There are several mechanisms by which dysfunctional telomeres result in telomere end fusions leading to chromosomal/genomic instability. For example, Shelterin provides protection against telomere fusions by telomere capping [5], and excessively shortened telomeres or base damage in telomeres inevitably results in reduced Shelterin occupancy [60]. Furthermore, generation of ROS due to mitochondrial dysfunction can damage telomeres, leading to accelerated telomere attrition by modifying guanines to 8-oxoguanines, which, if left unrepaired, can also lead to accelerated telomere loss and generation of NPBs [61,62]. It has been shown that oxidative stress plays an important role in prostate cancer [63,64]. Therefore, it is possible that all the events mentioned here collectively could lead to a significantly elevated frequency of NPBs and telomere shortening.

It is also relevant to consider potential confounding factors that may influence the CBMN assay results. For example, a recent study by Gajski et al. [65] reported that the frequency of MN and NPBs may be influenced by age, gender, radiation exposure and sampling season. In our study, there were no significant differences in age in controls and PC cases and all subjects were males; furthermore, there were no differences in sampling season because cases and controls were recruited continuously and simultaneously over a period of 34 months. There were no differences between cases and controls regarding lymphocyte MN frequency, which is one of the best validated biomarkers used for radiation exposure biodosimetry [66,67]; this suggests that any differences between groups in exposure to ionizing radiation were not sufficiently high enough to be a significant confounding factor in our study.

A potential weakness of our case–control study is the possibility of reverse causality, i.e., that having PC may somehow induce NPBs and telomere shortening in white blood cells. To overcome this limitation, it will be necessary to conduct expensive prospective studies to determine the following: (i) increased NPBs and/or shorter TL in apparently healthy younger men identifies those who eventually succumb to severe PC later in life and (ii) dietary and lifestyle interventions that reduce NPBs and telomere shortening prevent or delay the initiation and/or progression of PC. To perform this research successfully will also require a concerted effort to fully understand which nutritional, lifestyle, environmental and genetic factors may predispose men to elevated NPB frequency and/or telomere shortening.

## 5. Conclusions

The results obtained in the present study indicate that telomere-shortening-dependent chromosome instability, especially in advanced stages of the disease, is mediated by NPBs, and shorter TL and increased NPBs are associated with higher Gleason scores. In conclusion, (i) NPBs and TL can identify PC cases at higher risk of severe disease, (ii) it may be worthwhile to investigate in future studies if telomere shortening and/or NPBs formation in specific chromosomes may be more strongly involved in PC risk and/or progression and (iii) understanding which dietary and lifestyle factors increase the risk of NPB formation and telomere shortening in PC may contribute to new strategies for the prevention of PC.

## Figures and Tables

**Figure 1 cancers-15-03351-f001:**
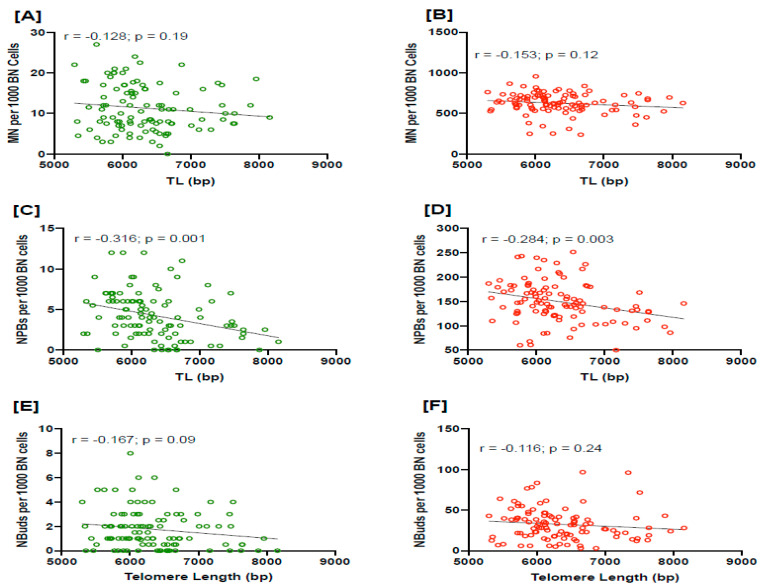
The correlation of baseline and radiation-induced DNA damage biomarkers MN (**A**,**B**), NPBs (**C**,**D**) and NBuds (**E**,**F**), respectively, with telomere length. MN, micronuclei; NPBs, nucleoplasmic bridges; NBuds, nuclear buds; BN cells, binucleated cells; bp, base pairs.

**Figure 2 cancers-15-03351-f002:**
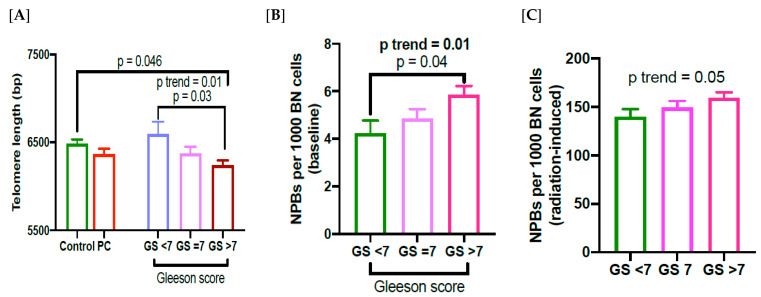
(**A**) Telomere length (TL) in healthy controls and PC cases (left panel), and TL in three different stages of cancer based on the Gleason score (right panel). (**B**,**C**) frequency of baseline and radiation-induced NPBs in different stages of disease stratified based on Gleason score. MN, micronuclei; NPBs, nucleoplasmic bridges; NBuds, nuclear buds; BN cells, binucleated cells; GS, Gleason score.

**Figure 3 cancers-15-03351-f003:**
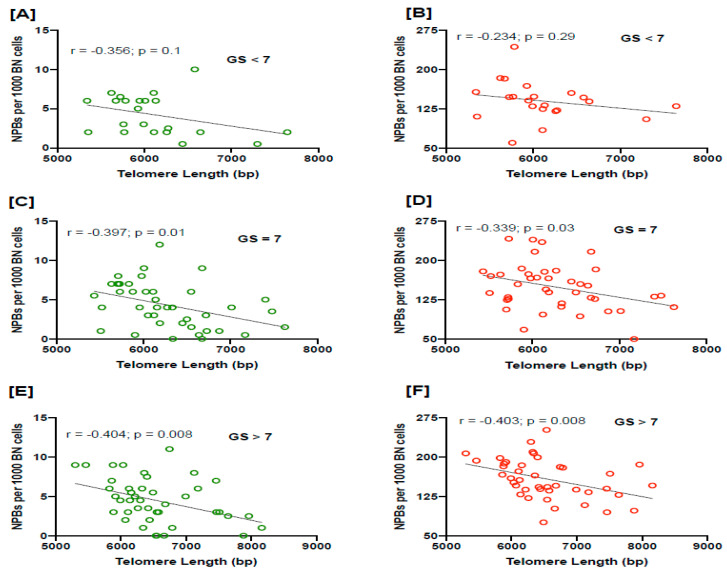
The correlation of baseline and radiation-induced NPBs with telomere length in different stages of PC stratified as per Gleason score: <7 (**A**,**B**), 7 (**C**,**D**) and >7 (**E**,**F**), respectively. MN, micronuclei; NPBs, nucleoplasmic bridges; NBuds, nuclear buds; BN cells, binucleated cells; bp, base pairs.

**Figure 4 cancers-15-03351-f004:**
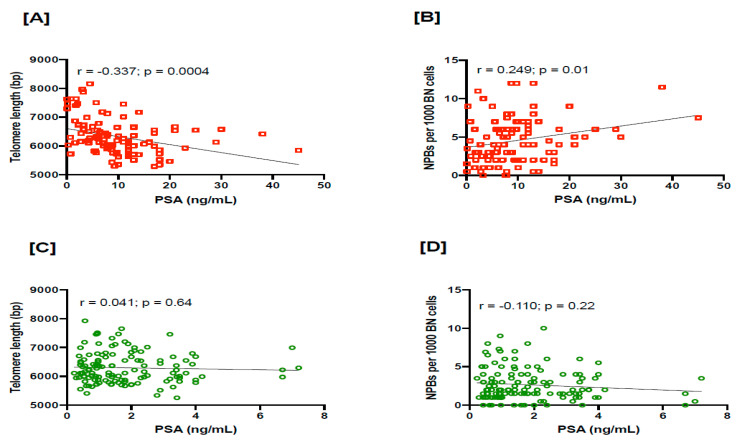
The correlation in PC cases of (**A**) PSA with TL and (**B**) PSA with NPBs. The correlation in controls of (**C**) PSA with TL and (**D**) PSA with NPBs. PSA, prostate-specific antigen; PC, prostate cancer; TL, telomere length; NPBs, nucleoplasmic bridges.

**Figure 5 cancers-15-03351-f005:**
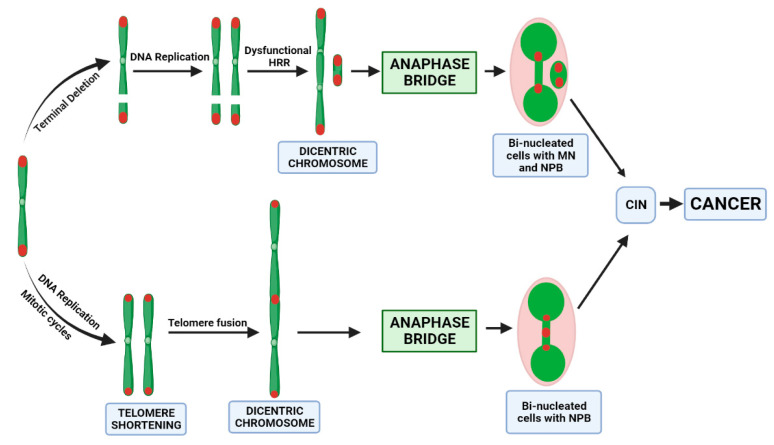
Proposed mechanisms by which telomere loss or telomere shortening induces nucleoplasmic bridge (NBP) formation, which leads to chromosomal instability (CIN) and an ultimately higher cancer risk.

**Table 1 cancers-15-03351-t001:** Comparison of prostate cancer cases and controls by selected demographic and clinical variables.

Characteristics	Cases	Controls	*p* Value
Age (years; mean ± SD)	71.24 ± 7.18	69.07 ± 7.99	0.88
PSA (ng/mL; mean ± SD)	9.5 ± 8.5	2.4 ± 2.45	0.0001
Smoking status			
Current smokers	9	3	0.0001
Ex-smokers	60	39	
Non-smokers	25	54	
Un-declared	12	36	
Gleason score (GS)			
(<7)	23	-	
(7)	41	-	
(>7)	42	-	
Telomere length (TL; bp)	6365 ± 648.1	6454 ± 569.8	0.26
TL and GS			
(<7)	6595 ± 675.9		
(7)	6373 ± 484.7		
(>7)	6241 ± 438.2		0.03
MN (baseline)	11.64 ± 0.62	10.70 ± 0.48	0.23
MN (radiation-induced)	641 ± 10.65	653.30 ± 8.3	0.62
NPBs (baseline)	4.26 ± 0.28	2.72 ± 0.18	0.0001
NPBs (radiation-induced)	151.8 ± 4.27	149 ± 2.86	0.62
NBuds (baseline)	1.75 ± 0.16	1.35 ± 0.14	0.057
NBuds (radiation-induced)	33.99 ± 2.29	23.97 ± 1.75	0.0005

## Data Availability

Data will be uploaded to a publicly available repository upon acceptance of the manuscript.

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
