# Peer review of "The Relationship between Telomere Length and Nucleoplasmic Bridges and Severity of Disease in Prostate Cancer Patients"

_cancers, 2023, doi:10.3390/cancers15133351_

Round 1

Reviewer 1 Report

The manuscript entitled “The relationship between telomere length and nucleoplasmic bridges in prostate cancer patients differs from that of healthy controls” aimed to investigate if telomere length is shorter in prostate cancer patients compared to healthy age-matched individuals. Based on the obtained results, authors suggest that nucleoplasmic bridges and telomere length might identify prostate cancer cases at higher risk of severe disease which could be useful in clinical setting and thus deserves publication in your respected Journal.

In the Introduction section, when talking about the risk of cancer it could be worth to mentioned also the increased cancer costs and other forgotten public health impacts of cancer. Please see:

Viegas et al. Forgotten public health impacts of cancer - an overview. Arh Hig Rada Toksikol. 2017; 68(4): 287-297. doi: 10.1515/aiht-2017-68-3005.

Authors could briefly acknowledge other biological effects of micronuclei in line with the possibility that fragmented DNA in the form of micronuclei exposed to the cytoplasm may trigger the activation of immunity-system-related genes. Please see:

Gekara, N.O. DNA damage-induced immune response: Micronuclei provide key platform. J. Cell Biol. 2017, 216, 2999–3001.

Under the paragraph Study participants, I would suggest to include some information regarding Ethics procedure that is already nicely indicated at the end of the paper.

Although it is stated that detailed description about the study population, etc. are described in authors previous publications it would be worth to provide some brief information on the study populations in this paragraph as well.

Authors could acknowledge some other factors that can confound the results of micronucleus assay such as seasonal variations. Please provide data, if possible, in which season the study population was collected.

Gajski et al. Cytokinesis-block micronucleus cytome assay parameters in peripheral blood lymphocytes of the general population: Contribution of age, sex, seasonal variations and lifestyle factors. Ecotoxicol Environ Saf. 2018; 148: 561-570. doi: 10.1016/j.ecoenv.2017.11.003.

Authors are also encouraged to discus in more detail possible clinical applications of their findings.

Minor remarks:

Page 2, line 73 – please change to “Hence the present study was carried out to test the hypotheses that (i) telomere length…”

Page 7, line 294 – there is no need for the abbreviation GI since it is also mentioned once.

In Figures some abbreviations have full word and some not and vice versa. Maybe to unify that?

Reviewer 2 Report

1. Did patients in the control group have any signs of benign prostatic hyperplasia? chronic prostatitis?

2. Why are patients with prostate cancer not described on the TNM scale? Did you have metastases in lymph nodes, as well as distant metastases, including in the bones? It is necessary to add information about patients and make calculations taking into account additional data.

3 Has there been a correlation between PCA levels and telomere length? The Glisson index should correlate with PCA, so a correlation with telomere length can also take place.

Round 2

Reviewer 2 Report

I have no more comments on the article. I believe that in its present form the manuscript can be recommended for publication.